# Autoantibodies Associated with Autoimmune Liver Diseases in a Healthy Population: Evaluation of a Commercial Immunoblot Test

**DOI:** 10.3390/diagnostics12071572

**Published:** 2022-06-28

**Authors:** Awais Ahmad, Charlotte Dahle, Johan Rönnelid, Christopher Sjöwall, Stergios Kechagias

**Affiliations:** 1Department of Biomedical and Clinical Sciences, Division of Inflammation and Infection/Clinical Immunology & Transfusion Medicine, Linköping University, SE-581 83 Linköping, Sweden; charlotte.dahle@regionostergotland.se; 2Department of Immunology, Genetics and Pathology, Uppsala University, SE-752 36 Uppsala, Sweden; johan.ronnelid@igp.uu.se; 3Department of Biomedical and Clinical Sciences, Division of Inflammation and Infection/Rheumatology, Linköping University, SE-581 83 Linköping, Sweden; christopher.sjowall@liu.se; 4Department of Health, Medicine and Caring Sciences, Division of Diagnostics and Specialist Medicine/Gastroenterology & Hepatology, Linköping University, SE-581 83 Linköping, Sweden; stergios.kechagias@liu.se

**Keywords:** autoimmune liver disease, AIH, PBC, PML, Sp100, gp210, AMA-M2, EUROLINE

## Abstract

Autoantibodies constitute important tools for diagnosing the autoimmune liver diseases (AILD) autoimmune hepatitis and primary biliary cholangitis. The EUROLINE immunoblot assay, detecting multiple specificities, is widely used, but the clinical importance of weakly positive findings is unclear. The manufacturer’s recommended cut-off was evaluated by investigating AILD-associated autoantibodies in 825 blood donors and 60 confirmed AILD cases. Positive findings were followed up with immunofluorescence microscopy on rat tissue, anti-M2-ELISA, alternative immunoblot assay, and liver function tests. Thirty-six (4.4%) blood donors were positive with EUROLINE. The most common specificities were LC-1 (1.6%), gp210 (1.3%), and AMA-M2 (1.1%). In general, the positive results were higher in patients than in blood donors, whereas anti-LC-1 was higher in blood donors. The liver function tests were slightly elevated in 2 of the 36 immunoblot positive blood donors. The majority of the positive EUROLINE findings could not be confirmed with the follow-up tests. The EUROLINE-Autoimmune Liver Diseases-(IgG) immunoblot detected autoantibodies in 4.4% of blood donors without signs of AILD. Our findings indicate that the recommended cut-off can be raised for most specificities without loss of diagnostic sensitivity. The prevalence of anti-LC-1 among blood donors indicates a problem with the antigen source.

## 1. Introduction

Several autoantibodies are important diagnostic markers for the autoimmune liver diseases (AILD) autoimmune hepatitis (AIH) and primary biliary cholangitis (PBC) [1,2,3]. The reported prevalence of AIH and PBC varies between 2–40 per 100,000. However, the correct number is difficult to establish due to undetected cases with mild or no symptoms [4,5]. Early diagnosis of AILD is important, since adequate treatment can prevent the development of end-stage liver disease. Without treatment, a mortality rate of 50% within the first 5 years after diagnosis of AIH has been reported [6,7]. In PBC, the disease course is usually indolent for many years, but with time, the destruction of bile ducts may result in pronounced cholestasis with the development of cirrhosis and portal hypertension [2,8]. 

The diagnosis of AIH is based on several criteria, including positive serology with anti-nuclear antibodies (ANA), smooth muscle antibodies (SMA), antibodies against liver-kidney-microsomal antigen (LKM), and soluble liver antigen (SLA) [9,10]. Several of these autoantibodies have high diagnostic specificity, and testing for their presence is performed when the etiology of abnormal liver function tests is unknown.

In PBC, anti-mitochondrial antibodies of M2-type (AMA-M2) are strong markers for the present or future development of the disease and can be detected in about 90–95% of cases [8]. AMA-M2 have high diagnostic specificity and the prevalence in healthy individuals is reported to be low (0.5–0.64%) [11,12]. In addition to AMA-M2, several subgroups of ANA have high diagnostic specificity for PBC but are less frequently found [5]. 

For decades, indirect immunofluorescence (IIF) microscopy on tissue sections of rat stomach, kidney, and liver has been the gold standard for the detection of SMA, AMA, and anti-LKM [13,14,15]. However, in line with the identification of specific target antigens, several immunoassays using purified or recombinant antigens have become commercially available and are today widely used [5,16]. These tests, e.g., immunoblot, may have a high sensitivity, although the clinical significance of weakly positive findings is unknown. 

Regardless of the technique used for the detection of autoantibodies, an optimal cut-off that is balanced between sensitivity and specificity is crucial. Manufacturers of commercial tests often recommend a cut-off corresponding to the 95th percentile, which implies that a false positive rate of 5% is accepted. However, if this cut-off is applied for tests that are widely used for screening of diseases with a low prevalence, the number of positive results, that may be clinically irrelevant, will outnumber the true positive cases.

An example is AMA-M2, which when detected by IIF is regarded to have a high diagnostic specificity for PBC, whereas the clinical relevance of weakly positive results with sensitive antigen-specific tests is unknown [17]. Weakly positive results may thus lead to the ineffective use of health care resources since autoantibody-positive cases often undergo further investigations. In the case of positive AMA-M2 with serum alkaline phosphatase (ALP) within reference limits, the diagnosis of PBC cannot be established. However, in these cases the 5-year incidence of PBC was reported to be 16% and, therefore, it seems reasonable to follow these patients at 2–3-year intervals [18,19]. 

A strict cut-off based on the evaluation of samples from individuals with liver function tests within reference limits is therefore warranted. The manufacturer’s recommended cut-off is usually based on the analyses of a limited number of sera from healthy individuals and the rule is that each laboratory should verify it with samples from the local population before taking the assay into use. However, these antigen-specific tests are expensive and larger studies of healthy populations are usually not performed. In addition, it is very important to verify the cut-off with samples from a large number of cases with confirmed diagnosis. However, for a clinical laboratory, it is difficult to obtain such samples, since the diagnosis is often unclear when samples are sent for testing in early stages of the disease. Furthermore, clinical information is often lacking at the time of request. The biobanking of samples from individuals with a confirmed diagnosis and close collaborations with treating physicians are therefore very important to obtain samples from true positive cases.

The main aim of this study was to evaluate the prevalence of AILD-associated autoantibodies detected with a commercial immunoblot in a large cohort of blood donors using the manufacturers’ recommended cut-off and to relate the autoantibody findings to liver function tests. Sera with positive autoantibody findings were further analyzed with additional antigen-specific assays, as well as with IIF microscopy on HEp-2 cells and rat tissue. In addition, the immunoblot results were compared with the corresponding findings in patients with confirmed PBC or AIH.

## 2. Materials and Methods

### 2.1. Blood Donors

825 blood donors, comprising of 403 females (48.8%) with a median age of 46 years (range 18–71) and 422 (51.2%) males with a median age of 42 years (range 19–77), were recruited from one blood donation center in Linköping, Sweden, during a two-year period (March 2018 to June 2019). Three individuals (0.4%) were ≥70 years of age (Table 1). 

Blood sampling was performed in conjunction with routine blood donation, and sera were stored at −70 °C until analysis. 

### 2.2. The AILD Cohort

This cohort comprised 60 patients diagnosed with AILD between 2006 and 2020 at the Department of Gastroenterology, The University Hospital, Linköping, Sweden. Of the 60 patients, 43 (71.7%) were females. Moreover, 30 (50%) patients had AIH according to the simplified criteria for the diagnosis of AIH, 27 (45%) had PBC according to criteria in the 2018 PBC guidelines, and 3 (5%) were diagnosed with AIH–PBC overlap syndrome (Table 1) [18,20].

Clinical data and autoantibody test results were retrieved from medical records. The autoantibody tests were performed at the time of diagnosis and analyzed in the same laboratory using the EUROLINE-Autoimmune liver diseases-(IgG) immunoblot assay in the same way as with the blood donor sera. 

### 2.3. Autoantibody Analysis

A commercial immunoblot assay (EUROLINE-Autoimmune Liver Diseases-(IgG); Euroimmun, Lübeck, Germany) was used according to the manufacturer’s instructions for detection of AILD-specific autoantibodies against M2 (i.e., E2 subunit of pyruvate dehydrogenase); M2-3E (BPO, i.e., a recombinant fusion protein of the E2-subunits of the three main M2-antigens; branched-chain 2-oxoacid dehydrogenase complex [BCOADC]; pyruvate dehydrogenase complex [PDC]; and 2-oxoglutarate dehydrogenase complex [OGDC]); speckled protein 100 (Sp100); promyelocytic leukaemia protein (PML); glycoprotein 210 (gp210); liver kidney microsomal 1 (LKM-1) antigen; liver cytosol 1 (LC-1) antigen; soluble liver antigen (SLA); and SSA/Ro52. The samples were analysed with EUROBlotmaster (Euroimmun AG, Euroimmun Lübeck, Germany), and after drying, the strips signal intensities were evaluated with EUROLineScan. The manufacturer’s recommended cut-off was used, i.e., signal intensity (grey scale units) >10 for all specificities. According to the Food and Drug Administration (FDA) 510(k) Substantial Equivalence Determination (k113439), a sample is positive if the respective band is clearly visible. 

A signal intensity between 6–10, defined by the manufacturer as borderline, was considered negative in our study. A signal intensity between 11–25 and 26–50 is considered positive with a medium to strong band. Values >50 result in a very strong band and are considered strongly positive. We defined low signal intensity as values <25.

The M2 antigen is natively purified from bovine heart. The remaining antigens are all recombinantly produced. 

PBC-associated autoantibodies were defined as AMA-M2, autoantibodies against BPO, Sp100, gp210, PML or PBC-associated IIF ANA-patterns (nuclear envelope, AC-11 and/or multiple nuclear dots, AC-6). The designation AMA includes both AMA-M2 and anti-BPO.

AIH-associated autoantibodies were defined as SMA, autoantibodies against SLA, LKM-1 or LC-1. Isolated findings of anti-SSA/Ro52 were not included as it is also frequently detected in several other autoimmune diseases.

### 2.4. Internal Controls

A pooled internal control with high signal intensity values regarding AMA-M2, BPO, Sp100, PML, gp210, LKM-1 and SLA, has been used in our laboratory since 2014. The coefficient of variation (CV) values for these specificities were retrieved for the period between 2018–2019, during which the blood donor sera were analyzed. CV for lower signal intensity levels was evaluated by diluting the pooled internal control and analyzed with EUROLINE immunoblot on five occasions with one-week intervals. Anti-LC-1 was not included in the pooled internal control due to lack of suitable positive sera.

Mean signal intensity for our pooled internal control with high signal intensity values ranged between 36–145 (SD 5–15). CV for each specificity was: AMA-M2 (11%), BPO (15%), Sp100 (7%), gp210 (6.5%), PML (17%), LKM-1 (14%) and SLA (3%). 

The mean signal intensity values for the specificities with low signal intensity values ranged 15–25 (SD 1–5). CV for these specificities were: AMA-M2 (8.2%), BPO (10.3%), Sp100 (8.8%), gp210 (5.9%) and LKM-1 (22.7%).

### 2.5. EliA, Alegria^®^ and Liver-9-Line

Blood donor sera with any positive EUROLINE immunoblot specificity of AMA-M2, BPO, gp210, Sp100, LC-1 and SLA/LP were also analysed for these specificities with ELISA (Alegria^®^, Mainz, Germany) and an immunoblot assay (Liver-9-line 2nd generation), both from Orgentec Diagnostika GmbH (Mainz, Germany), in order to verify the results. In addition, blood donor sera with EUROLINE AMA-M2 and/or anti-BPO positivity were analysed with fluorescence enzyme immunoassay (EliA™ Phadia 250, Thermo Fisher Scientific, Phadia AB, Uppsala, Sweden). The manufacturer’s recommended cut-off (>10) was used, and borderline results (6–10) were considered negative.

### 2.6. IIF ANA

ANA were detected by IIF microscopy on HEp2-cells (Immunoconcepts, Sacramento, California, USA) as previously described [21]. With a screening dilution of 1:800, the cut-off for a positive ANA corresponds to the 95th percentile among healthy blood donors (*n* = 752; 50% females, 50% males), in agreement with the international recommendations [13]. 

### 2.7. AMA, SMA, and Anti-LKM with IIF Microscopy

SMA, AMA, and anti-LKM were detected by IIF microscopy on slides with sections of fixed rat liver, kidney, and stomach (NOVA Lite^®^, Werfen, San Diego, CA, USA). For the detection of SMA, the slides were incubated for 30 min with serum diluted 1:400 in phosphate buffered saline (PBS). This serum dilution corresponds to the 95th percentile of positive SMA in healthy blood donor sera (*n* = 200). After washing and 30 min incubation with FITC-conjugated γ-chain-specific anti-human IgG (Werfen) the slides were mounted in fluorescent mounting medium (DAKO, Glostrup, Denmark) and evaluated by IIF microscopy (Olympus BX43, Olympus, Tokyo, Japan) at 10× magnification. Blood donor sera that were positive for AMA-M2 or anti-LKM-1 on immunoblot, but negative when analysed on rat-tissue in serum dilution 1:400, were also analysed at dilution 1:40. All autoantibody tests were performed at the accredited laboratory of Clinical Immunology, Linköping, Sweden.

### 2.8. Routine Laboratory Measurements

Alkaline phosphatase (ALP), alanine aminotransferase (ALT), aspartate aminotransferase (AST), and gamma-glutamyl transferase (GGT) were analysed with a spectrophotometric technique according to routine methods at the accredited laboratory of Clinical Chemistry, University Hospital in Linköping (Cobas c 701, Cobas^®^ 8000, Roche Diagnostics International Ltd., Rotkreuz, Switzerland).

### 2.9. Statistics

For comparison of autoantibody signal intensity values between AILD and blood donors, Student’s t-test was used. *p*-values < 0.05 were considered significant. Sensitivity and specificity were evaluated with Receiver operating characteristic (ROC) analysis. The software used was Microsoft Excel^®^ version 2018 (Redmond, WA, USA).

## 3. Results

### 3.1. Blood Donor Cohort

#### 3.1.1. EUROLINE

In total, sera from 36 individuals (4.4%) were positive, out of which 16 (44.4%) were females. In 35 sera, the autoantibody findings were isolated, and 21 of these sera had low signal intensity values—see Figure 1. Serum from one subject was positive with multiple PBC-associated autoantibodies with signal intensity values ranging from 24 to 50 without any laboratory signs of cholestasis.

In total, the most frequent finding was anti-LC-1 that was found in 13 (1.6%) of the 825 blood donors. Another 11 individuals (1.3%) tested positive for anti-gp210, 9 (1.1%) for AMA-M2 and/or anti-BPO, and 3 for anti-Sp100 (0.3%)—see Figure 2 and Table 2.

One individual, a male, was weakly positive for anti-SLA (signal intensity 12) but had normal serum liver tests. Anti-PML or anti-LKM-1 were not detected in any of the blood donor sera.

No differences were found between males and females regarding the different specificities. 

#### 3.1.2. EliA, Alegria^®^ and Liver-9-Line

Out of the nine sera that were EUROLINE AMA-M2 and/or anti-BPO positive, four (44.4%) were also AMA-M2 positive with EliA™, two (22.2%) with Alegria^®^ and one (11.1%) with Liver-9-line assay. Two of the EUROLINE AMA-M2-positive sera (with signal intensity 72 and 35, respectively) were weakly positive for AMA with IIF-microscopy, and these were also positive with Alegria^®^ and EliA™. One of these two IIF-positive sera was weakly positive for AMA-M2 with Liver-9-line. None of these had signs of cholestasis.

All three EUROLINE anti-Sp100 positive sera were verified with Liver-9-line and one of them had positive ANA with nuclear dots pattern (AC-6). Two of these three sera were also analysed and were positive with Alegria^®^, whereas the third could not be analysed due to lack of serum. Out of the EUROLINE anti-LC-1 positive sera (*n* = 13), two (15.4%) could be verified with Line-9-line assay and one (7.7%) with Alegria^®^. The weakly positive EUROLINE anti-SLA/LP serum was also weakly positive with Liver-9-line assay, but negative with Alegria^®^. None of the EUROLINE anti-gp210 positive sera (*n* = 11) could be verified with Alegria^®^ or the Line-9-line assay, and only one had a weakly positive ANA with homogeneous pattern (AC-1).

#### 3.1.3. IIF Microscopy

ANA were detected in 61 (38 females, 23 males) out of the 825 blood donors (7.4%). Two females had the PBC-associated ANA pattern nuclear dots (AC-6) with specificity for Sp100, and one of them also had the cytoplasmic AMA-pattern (AC-21) on HEp-2 cells. 

In total, seven individuals (0.8%) were positive with IIF microscopy on rat tissue. Out of these, the two females with nuclear dots ANA (AC-6) mentioned above were weakly positive for AMA and specificity for AMA-M2 and BPO could be verified. The remaining five were males and were weakly SMA-positive. 

#### 3.1.4. Serum Liver Tests

Out of the 36 EUROLINE positive blood donors, two had slightly elevated serum liver tests; one female with isolated anti-Sp100 positivity had ALP 117.6 U/L (ref 42–114 U/L) and one male with isolated anti-LC-1 positivity had AST 69 (ref < 46 U/L). Three males with isolated SMA also had slight elevations of AST or GGT. All of them had normal ALT levels. None of them were diagnosed with AILD.

### 3.2. AILD Patients

#### 3.2.1. EUROLINE

A total of 10 (33.3%) out of the 30 AIH-patients were EUROLINE immunoblot-positive. The most frequent findings were anti-SLA (*n* = 6, 20%), AMA-M2 and/or anti-BPO (*n* = 3, 10%), anti-LC-1 (*n* = 2, 6.7%), and anti-gp210 (*n* = 2, 6.7%)—see Table 2. None of these patients were anti-PML or anti-LKM-1-positive. 

Moreover, 26 (96.3%) out of the 27 PBC-patients were EUROLINE immunoblot-positive; AMA-M2 (*n* = 22, 81.5%), anti-BPO (*n* = 24, 88.9%), anti-Sp100 (*n* = 8, 29.6%), anti-gp210 (*n* = 5, 18.5%), and anti-PML (*n* = 2, 7.4%)—see Figure 3 and Table 2. AMA-M2 and anti-BPO were concomitantly positive in 24 (88.9%) of the PBC cases. One of the AMA positive patients was also anti-SLA positive.

In the AIH/PBC overlap cohort, two patients had anti-SLA, out of which one had AMA-M2 and anti-BPO and the other had anti-gp210. The third had AMA-M2 and anti-BPO—see Table 2.

#### 3.2.2. IIF Microscopy

A total of 18 (62.1%) patients in the AIH group were ANA positive, out of which 9 with a homogeneous pattern (AC-1). Moreover, 20 out of the 30 AIH patients (66.7%) were positive with IIF microscopy on rat tissue and 18 (60%) of them were SMA-positive. Two females (6.7%) were AMA positive. With EUROLINE immunoblot, one of them had anti-SLA (signal intensity 46) and the other had a weakly positive anti-LC-1 (signal intensity 20). No AIH-patient was anti-LKM-1 positive.

Seven (25.9%) of the twenty-seven PBC patients were ANA positive, and five of them had nuclear dots pattern (AC-6). AMA was detected on IIF rat tissue in 25 (92.6%) patients. Out of these, 24 (96%) were also positive with immunoblot (AMA-M2 and/or anti-BPO). Two patients (7.4%) were SMA positive, one of which was also AMA-positive. Two patients (7.4%) were AMA negative. Of these, one was isolated anti-Sp100 positive, and the other was positive for anti-Sp100, anti-gp210, anti-PML and SMA. 

One patient in the AIH/PBC overlap cohort was ANA positive and had a homogeneous pattern (AC-1). The three patients were all positive on IIF rat tissue. Two were AMA positive, of which one also was SMA positive. The third patient was SMA positive.

### 3.3. Comparison of EUROLINE Immunoblot Results in AILD Patients and Blood Donors

In total, the mean signal intensity values of positive samples were about three times higher in the patient group (*p* < 0.001)—see Figure 1 and Table 2. In contrast, anti-LC-1 positive blood donors (*n* = 13) had higher signal intensity values compared with the anti-LC-1 positive AILD patients (*n* = 2).

### 3.4. Sensitivity and Specificity

#### 3.4.1. NIBSC Reference Serum 67/183

Signal intensity values with EUROLINE’s immunoblot for AMA-M2 and anti-BPO in the dilutions corresponding to 50, 10, 5, and 1 U/mL were 119, 104, 92, and 77, respectively, for AMA-M2 and 103, 67, 51, and 38, respectively, for anti-BPO. With EliA™, the corresponding dilutions resulted in 89, 39, 23, and 8 U/mL (cut-off > 6). The dilutions corresponding to 10 and 20 U/mL were weakly positive for AMA with IIF microscopy, whereas the lower concentrations were all negative in serum dilution 1/40. 

#### 3.4.2. Cut-Off

With the manufacturer’s recommended cut-off (>10), AMA-M2, BPO, Sp100, gp210, and SLA all have a specificity >98% (99.2, 99.4, 99.6, 98.7, and 99.9%, respectively)—see Figure 4.

Raising the cut-off for AMA-M2 from >10 to >27 increases the specificity from 99.2 to 99.5% without loss of sensitivity. Similarly, the sensitivity is not affected by raising the cut-off for anti-BPO from >10 to >16. 

AMA-M2 and anti-BPO were concomitantly positive in 24 (88.9%) of the PBC cases—see Figure 3. By combining the results, the cut-off of either AMA-M2 or anti-BPO can be raised to >41 with retained sensitivity and results in increased specificity (99.6%).

Raising the cut-off >13 for anti-Sp100 and >24 for anti-gp210, respectively, does not result in reduced sensitivity. Regarding anti-SLA, the cut-off can be raised to >15 without decreased sensitivity.

## 4. Discussion

This study shows that the widely used commercial immunoblot EUROLINE-Autoimmune Liver Diseases-(IgG) detects AILD-associated autoantibodies in 4.4% of a large blood donor cohort without being associated with elevated liver function tests. In most cases, the positive responses were isolated, weak, and could not be verified by alternative methods, including IIF. The most frequent positive specificities were LC-1, gp210, and AMA-M2 and/or anti-BPO. In contrast to the blood donor results, the positive results in patients with established AILD were, with few exceptions, significantly stronger. In total, we found a slightly higher prevalence of AMA (1.1%) compared to previous studies (0.5–0.64%) that used ELISA and IIF on rat kidney sections, respectively [11,12]. 

The EUROLINE immunoblot is highly sensitive for detection of AMA, since low concentrations of NIBSC PBC-reference serum as 1 U/mL resulted in strong positive reactions. This is in contrast to 10 U/mL corresponding to the lowest concentration of AMA that we could detect by IIF on rat tissue. 

To the best of our knowledge, studies on the prevalence of anti-gp210, anti-Sp100, and anti-LC-1 in healthy individuals are lacking.

Many of the autoantibody specificities detected by EUROLINE immunoblot are considered to be strong diagnostic markers for AIH and PBC. This implies that low positive findings may also be interpreted to support a diagnosis of AILD and thus lead to further investigations. However, since low positive results can also be found in healthy individuals, the diagnostic specificity of low levels detected by immunoblot seems to be low. 

In clinical routine, analyses of liver function tests are frequently requested, and elevated levels can have many underlying causes, of which AILD only constitute a minority. However, AILD are important to identify in early stages since treatments are available and can prevent the development of end-stage liver disease. The autoantibody tests are therefore widely used and, together with a panel of other tests, often included in the early stages of investigation when the etiology of elevated liver function tests is unknown. In this context, the pretest probability of AILD is low, and this underscores that the use of a test with a high diagnostic specificity is of utmost importance.

Our findings of a positive rate of 4.4% among blood donors implies that about one in twenty-five tested healthy individuals might be suspected to suffer from an AILD and be referred for further investigations. This risks causing unnecessary worries and ineffective use of health care resources.

Our findings indicate that the EUROLINE’s recommended cut-off is too low, implying a high risk for low positive results of uncertain clinical relevance. Since the prevalence of AILD is very low, EUROLINE cannot be used for population-based screening, even if the cut-offs are revised. 

Based on the information in the product insert of the EUROLINE immunoblot, it is not evident how the cut-off was validated or how an identical cut-off determined by densitometry can be used for all included antigens. According to the Food and Drug Administration (FDA) document 510(k) Substantial Equivalence Determination (number k113439), the assay cut-off is based on 261 clinically characterized samples from PBC and AIH patients and 171 control samples. The results are not further described, but it is stated that the negative and positive results could be clearly discriminated. In this document, it is also explained that a uniform cut-off for all antigens was obtained by identifying the optimal dilution for each antigen. This raises questions about the concentration of M2 antigen, since a very low concentration (1 U/mL) of NIBSC reference serum 67/183 resulted in high signal intensity values.

Furthermore, there also seems to be an issue with the recombinantly produced LC-1 antigen, since the signal intensity values were higher in blood donors than in the AILD group.

In our previous work, we found a high prevalence of AMA-M2 and/or anti-BPO (7.4%) in 272 SLE patients with the EUROLINE immunoblot [22]. However, most of them also had low signal intensity values, whereas six (2.2%) had a signal intensity >40. Even when using >40 as cut-off, the prevalence of AMA in SLE was still higher compared to the reported prevalence in healthy populations [11,12]. 

One weakness of our study is that the results of our AILD cohort were retrieved retrospectively; moreover, the AILD sample size was relatively small. On the other hand, EUROLINE has been used for many years in our laboratory and the internal controls have shown a stability over the years, supporting the validity of comparing the results with the more recently analyzed blood donor sera. Regarding positive reactions around the cut-off, the uncertainty may be greater, as the results can fluctuate between weakly positive and negative according to the previously mentioned FDA document. However, very few of our blood donor results had signal intensity values in close proximity to the cut-off.

Since most of our AILD patients (78.3%) were diagnosed within one year of sampling, we argue that medication has had little impact on the autoantibody results in this group. 

The strength of our study is the large blood donor cohort, the verification of findings with other methods, and the correlation of autoantibody findings to liver enzyme values. We plan to conduct a follow-up study in five years to evaluate whether the blood donors with positive autoantibody findings are prone to developing AILD. Our blood donor cohort included 257 females over 40 years of age and it would be of interest to perform a study focused on a larger cohort of this age group, since the incidence of PBC is highest in middle aged females.

## 5. Conclusions

Antigen-specific assays are important for the detection of AILD-associated autoantibodies and constitute important diagnostic tools when the cause of elevated liver enzyme values is unknown. Using panels that simultaneously detect multiple autoantibody specificities is advantageous, but it is important for the diagnostic specificity to be high. We analyzed sera from a large blood donor cohort with EUROLINE-Autoimmune Liver Diseases-(IgG) immunoblot and compared the results with patients with confirmed AIH and/or PBC. The positive results in the blood donors were mainly weak, could often not be verified by other techniques, and were not associated with elevated liver enzyme values. The finding that autoantibodies against LC-1 were more frequent and stronger positive among blood donors indicates a problem with the recombinant antigen source. Regarding AMA, a very low concentration of the NIBSC reference serum 67/183 resulted in a strong positive response with EUROLINE, supporting our suggestion that the cut-off should be adjusted. Our findings indicate that the manufacturer’s recommended cut-off can be raised for most specificities to increase the diagnostic specificity without loss of sensitivity. 

## Figures and Tables

**Figure 1 diagnostics-12-01572-f001:**
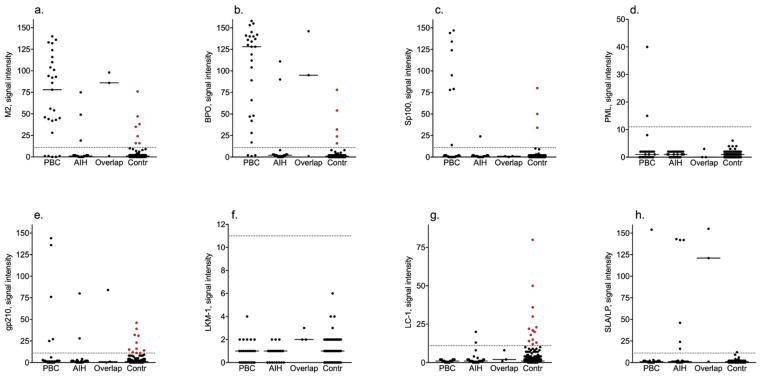
Signal intensity values of specificities in the blood donor and disease cohorts. Red dots indicate controls with signal intensity values above the manufacturer’s cut-off. PBC, primary biliary cholangitis; AIH, autoimmune hepatitis; Overlap, AIH-PBC overlap syndrome; Contr, control; (**a**) M2—E2 subunit of pyruvate dehydrogenase; (**b**) BPO—recombinant fusion protein of the E2-subunits of the three main M2-antigens branched-chain 2-oxoacid dehydrogenase complex [BCOADC], pyruvate dehydrogenase complex [PDC] and 2-oxoglutarate dehydrogenase complex [OGDC]; (**c**) Sp100—speckled protein 100; (**d**) PML—promyelocytic leukaemia protein; (**e**) gp210—glycoprotein 210; (**f**) LKM-1—liver kidney microsomal 1 antigen; (**g**) LC-1—liver cytosol 1 antigen; (**h**) SLA—soluble liver antigen.

**Figure 2 diagnostics-12-01572-f002:**
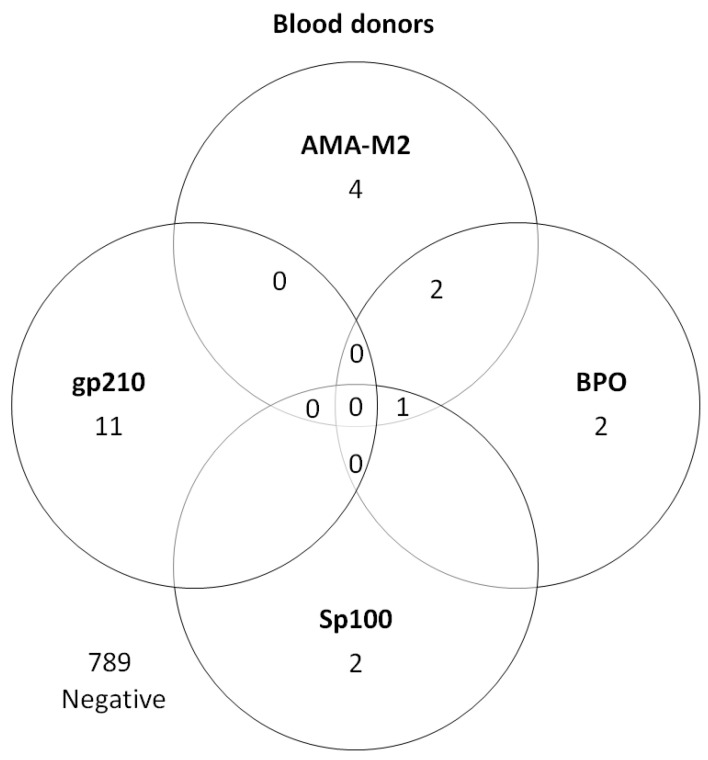
PBC-associated autoantibodies in blood donors (*n* = 22). AMA-M2—E2 subunit of pyruvate dehydrogenase; BPO—recombinant fusion protein of the E2-subunits of the three main M2-antigens branched-chain 2-oxoacid dehydrogenase complex [BCOADC], pyruvate dehydrogenase complex [PDC] and 2-oxoglutarate dehydrogenase complex [OGDC]; Sp100—speckled protein 100; gp210—glycoprotein 210.

**Figure 3 diagnostics-12-01572-f003:**
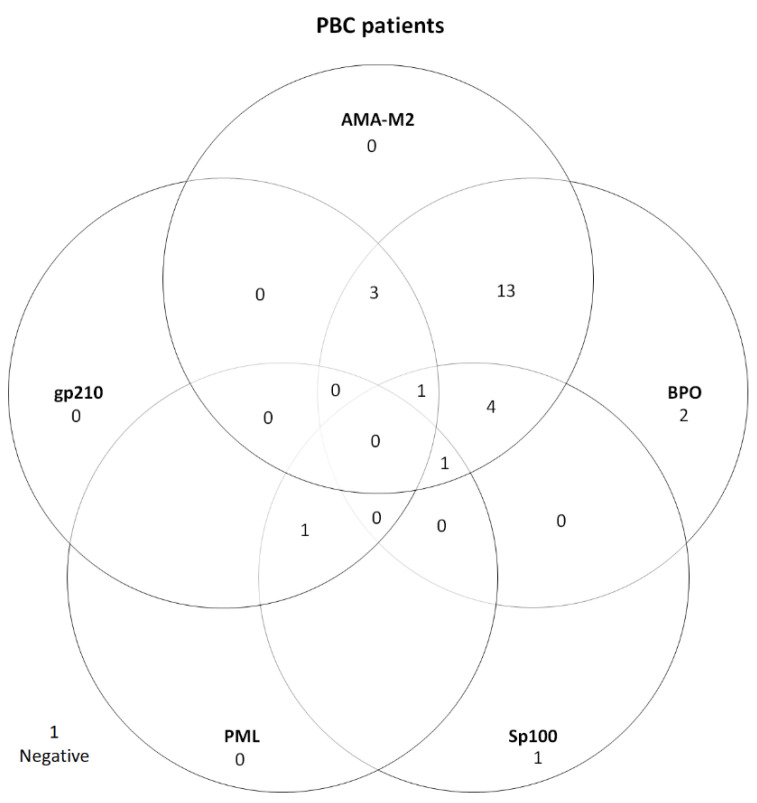
PBC-associated autoantibodies in PBC patients (*n* = 27). One PBC patient was negative with EUROLINE immunoblot but positive with IIF rat tissue technique. AMA-M2—E2 subunit of pyruvate dehydrogenase; BPO—recombinant fusion protein of the E2-subunits of the three main M2-antigens branched-chain 2-oxoacid dehydrogenase complex [BCOADC], pyruvate dehydrogenase complex [PDC] and 2-oxoglutarate dehydrogenase complex [OGDC]); Sp100—speckled protein 100; PML—promyelocytic leukaemia protein; gp210—glycoprotein 210.

**Figure 4 diagnostics-12-01572-f004:**
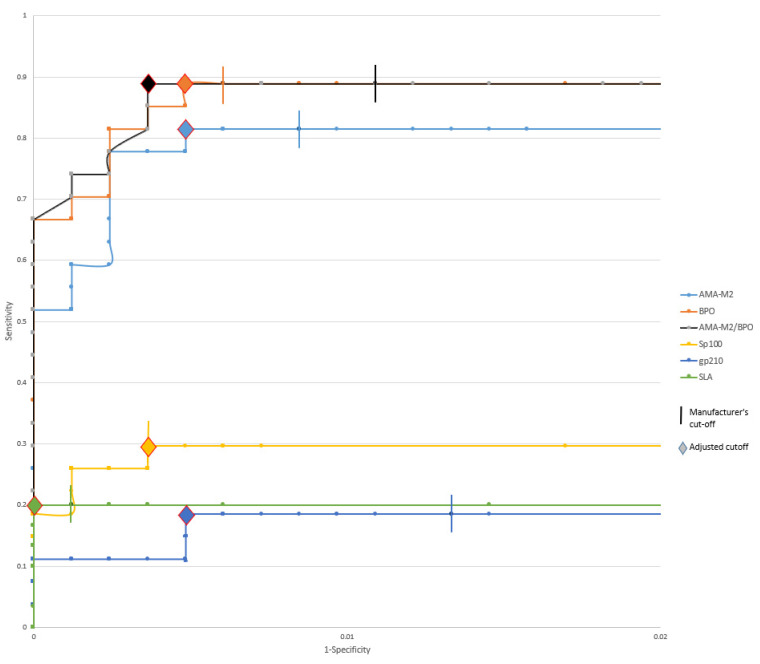
ROC curve for each specificity with EUROLINE’s recommended cut-off (>10) with truncated X-axis. AMA-M2—E2 subunit of pyruvate dehydrogenase; BPO—recombinant fusion protein of the E2-subunits of the three main M2-antigens branched-chain 2-oxoacid dehydrogenase complex [BCOADC], pyruvate dehydrogenase complex [PDC] and 2-oxoglutarate dehydrogenase complex [OGDC]); AMA-M2/BPO—AMA-M2 and/or BPO; Sp100—speckled protein 100; gp210—glycoprotein 210; SLA—soluble liver antigen.

**Table 1 diagnostics-12-01572-t001:** Demographics and duration of diagnosis.

	Blood Donors	AIH	PBC	AIH/PBC-Overlap
Total *n* (%)	825	30	27	3
Total median age (range)	43 (18–77)	56 (21–84)	65 (24–88)	46 (42–71)
Females *n* (%)	403 (48.)	19 (63.3)	21 (77.8)	3 (100)
Females median age (range)	46 (18–71)	60 (22–78)	62 (24–86)	46 (42–71)
Males *n* (%)	422 (51.2)	11 (36.7)	6 (22.2)	
Males median age (range)	42 (19–77)	38 (21–84)	75 (32–88)	
Duration of diagnosis (years)				
<0 ^a^ *n* (%)		3 (10)	6 (22.2)	0
0–1 *n* (%)		25 (83.3)	19 (70.4)	2 (66.7)
>1 *n* (%)		2 (6.7)	2 (7.4)	1 (33.3)

AIH autoimmune hepatitis; PBC primary biliary cholangitis; AIH/PBC overlap autoimmune hepatitis/primary biliary cholangitis overlap. ^a^ Indicates patients diagnosed during a five year period before first known year of sampling.

**Table 2 diagnostics-12-01572-t002:** Number of autoantibody-positive individuals, and the mean signal intensity of each specificity in the blood donor and disease cohorts.

Blood Donors (*n* = 36)	M2	BPO	Sp100	PML	gp210	LKM1	LC1	SLA/LP
*n*	7	5	3	0	11	0	13	1
Females (*n* = 16)	3	4	3		5		4	
Males (*n* = 20)	4	1			6		9	1
Median signal intensity (range)	35 (16–76)	32 (16–78)	50 (34–80)		16 (11–46)		21 (12–80)	12
AIH (*n* = 10)								
*n*	3	2	1	0	2	0	2	6
Females (*n* = 8)	3	2	1		1		2	5
Males (*n* = 2)					1			1
Median signal intensity (range)	49 (19–75)	101 (90–111)	24		54 (28–80)		12 (13–20)	86 (16–143)
PBC (*n* = 26)								
*n*	22	24	8	2	5	0	0	1
Females (*n* = 20)	17	18	6	2	4			1
Males (*n* = 6)	5	6	2		1			
Median signal intensity (range)	93 (28–140)	129 (17–158)	110 (14–147)	28 (15–40)	76 (25–144)			154
AIH/PBC overlap (*n* = 3, all females)								
*n*	2	2	0	0	1	0	0	2
Median signal intensity (range)	92 (86–98)	120 (95–146)			84			138 (121–155)

PBC, primary biliary cholangitis; AIH, autoimmune hepatitis; M2—E2 subunit of pyruvate dehydrogenase; BPO—recombinant fusion protein of the E2-subunits of the three main M2-antigens branched-chain 2-oxoacid dehydrogenase complex [BCOADC], pyruvate dehydrogenase complex [PDC] and 2-oxoglutarate dehydrogenase complex [OGDC]; Sp100—speckled protein 100; PML—promyelocytic leukaemia protein; gp210—glycoprotein 210; LKM-1—liver kidney microsomal 1 antigen; LC-1—liver cytosol 1 antigen; SLA—soluble liver antigen.

## Data Availability

Data available on request.

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
