# Peer review of "Autoantibodies Associated with Autoimmune Liver Diseases in a Healthy Population: Evaluation of a Commercial Immunoblot Test"

_diagnostics, 2022, doi:10.3390/diagnostics12071572_

Round 1
Reviewer 1 Report
The authors have evaluated EUROLINE immunoblot for AILD-associated autoantibodies in 825 blood donors and 60 confirmed AILD cases. This is a well written, interesting, and useful contribution, which I think is entirely suitable for publication in Journal of MDPI Diagnostics.
Major points:
1. Can authors indicate 26 positive individuals in different color in Figure 1?
2. Please explain the number outside the circles in Figure 2 a & b.
3. Please change the color of vertical lines which indicating manufacturer’s cut-off to respective color of horizontal lines for easier understanding.
Minor points:
1. Line 110: (table 1) should be (Table 1).
Author Response
Thank you for your positive response and constructive suggestions for improvement of the manuscript. The three major points as well as the minor point have now been adressed and changes have been made accordingly.
Best regards and thank you again
Awais Ahmad
Reviewer 2 Report
The study presented focuses on testing a commercial immunoblot test on a healthy population. The opportunity of performing such test is very interesting and as the authors mention, it may be helpful in early identification of patients.
I have some recommendations:
Please comment in the discussion section if screening programs are necessary if they should be embedded or not, and in what situations. Moreover, for better results would you recommend reproducing the study in a more focused population group, such as females between 40 to 60 years old?
Perhaps a flowchart might be useful to better show patient's inclusion in the study.
Author Response
Thank you for your positive response and constructive suggestions for improvement of the manuscript.
We have now added a part in the discussion about why these tests are not suitable for screening and that it would be desirable with a prospective study focused in females over 40 years of age.
We have chosen not to add a flowchart since we think that the information is provided in Table 1.
Best regards and thank you again
Awais Ahmad